# Short-Term Effects of Manual Therapy plus Capacitive and Resistive Electric Transfer Therapy in Individuals with Chronic Non-Specific Low Back Pain: A Randomized Clinical Trial Study

**DOI:** 10.3390/medicina59071275

**Published:** 2023-07-09

**Authors:** Konstantinos Kasimis, Paris Iakovidis, Dimitrios Lytras, Georgios Koutras, Ioanna P. Chatziprodromidou, Antonis Fetlis, Stefania Rafailia Ntinou, Natalia-Maria Keklikoglou, Antigoni Samiotaki, Georgios Chasapis, Georgia Tarfali, Thomas Apostolou

**Affiliations:** 1Laboratory of Biomechanics & Ergonomics, Department of Physiotherapy, Faculty of Health Sciences, International Hellenic University, Alexander Campus, P.O. Box 141, 57400 Thessaloniki, Greece; konstantinoskasimis@gmail.com (K.K.); paris_physio@yahoo.com (P.I.); kutrasg@otenet.gr (G.K.); antonisfetlis@gmail.com (A.F.); stefi_ntinou@hotmail.com (S.R.N.); nataliakeklikoglou12@gmail.com (N.-M.K.); antigonisamiotaki@gmail.com (A.S.); gxasapis@yahoo.gr (G.C.); apostolouthomas@hotmail.com (T.A.); 2Department of Public Health, Medical School, University of Patras, 26504 Patra, Greece; ioannachatzi@med.upatras.gr; 3Biomedical Engineering Department, Faculty of Engineering, Strathclyde University, Glasgow G4 0NW, UK; gtarfali@gmail.com

**Keywords:** chronic low back pain, manual therapy, capacitive and resistive electric transfer therapy

## Abstract

*Background and Objectives:* Chronic non-specific low back pain (CNSLBP) is defined as back pain that lasts longer than 12 weeks. Capacitive and resistive electric transfer (TECAR) therapy utilizes radiant energy to generate endogenous heat and is widely used for the treatment of chronic musculoskeletal pain. The aim of this study was to investigate the efficacy of manual therapy (MT) program combined with TECAR therapy in individuals with CNSLBP. *Materials and Methods:* Sixty adults with CNSLBP were randomly divided equally into three groups. The first group followed an MT protocol in the lumbar region (MT group), the second group followed the same MT protocol combined with TECAR therapy (MT + TECAR group) using a conventional capacitive electrode as well as a special resistive electrode bracelet, and the third group (control group) received no treatment. Both intervention programs included six treatments over two weeks. Pain in the last 24 h with the Numeric Pain Rating Scale (NPRS), functional ability with the Roland–Morris Disability Questionnaire (RMDQ), pressure pain threshold (PPT) in the lumbar region with pressure algometry, and mobility of the lumbo-pelvic region through fingertip-to-floor distance (FFD) test were evaluated before and after the intervention period with a one-month follow-up. Analysis of variance with repeated measures was applied. *Results:* In the NPRS score, both intervention groups showed statistically significant differences compared to the control group both during the second week and the one-month follow-up (*p* < 0.001). Between-group differences were also noticed between the two intervention groups in the second week (*p* < 0.05). Differences in the RMDQ score were detected between the intervention groups and the control group in the second week and at the one-month follow-up (*p* < 0.001), while differences between the two intervention groups were only detected at the one-month follow-up (*p* < 0.001). Regarding the PPT values, differences were found mainly between the MT + TECAR group and the control group and between the MT + TECAR group and the MT group (*p* < 0.05), with the MT + TECAR group in most cases showing the greatest improvement compared to the other two groups, which remained statistically significant at the one-month follow-up (*p* < 0.05). Finally, both intervention groups improved the mobility of the lumbo-pelvic region at both time points compared to the control group without, however, statistically significant differences between them (*p* > 0.05). *Conclusions:* The application of an MT protocol with TECAR therapy appeared more effective than conventional MT as well as compared to the control group in reducing pain and disability and improving PPT in individuals with CNSLBP. No further improvement was noted in the mobility of the lumbo-pelvic region by adding TECAR to the MT intervention.

## 1. Introduction

Chronic non-specific low back pain (CNSLBP) is defined as pain in the lower back that cannot be attributed to a known condition or etiology with symptoms that manifest for more than 12 weeks [1]. Lifetime prevalence of low back pain is 84% with 23% experiencing chronicity in their symptoms and 12% disability [2]. CNSLBP is typically accompanied by a reduced range of motion, long-lasting disability, and a lower quality of life, as well as psychosocial factors, such as anxiety and depression [3]. For more than three decades, low back pain has been the primary cause of disability worldwide, leading to significant expenses for healthcare and lost productivity [3,4]. Individuals who experience long-term disability due to chronic or recurring low back pain contribute most of the social and economic costs associated with this condition [5].

The matter has been extensively researched, but the optimal therapeutical approach remains to be decided. Manual therapies such as mobilization and manipulation of the lumbar area have provided positive results in dealing with CNSLBP, reducing pain, and improving disability [6,7,8,9]. Another approach which is widely used in clinical practice and seems to be gaining ground in recent years is the use of capacitive and resistive electric transfer (TECAR) therapy. Protocols with TECAR utilize radiant energy to produce internal heat, and they are used for musculoskeletal disorders due to their ability to relieve pain, relax muscles, and increase elasticity [10,11]. The two modes of TECAR therapy are resistive and capacitive, both of which are utilized for their therapeutic properties [12]. Research evidence suggests that the application of TECAR therapy on the lumbar area improves pain, disability, and range of motion in individuals with CNSLBP [12,13] as well as amplifies the therapeutic benefits of exercise [14]. Modern TECAR devices allow, with the use of special electrodes, the simultaneous application of TECAR with manual techniques or therapeutic exercise since they transform the hand of the physical therapist into an electrode, providing the possibility of a more dynamic treatment. One such electrode is the resistive electrode bracelet, which allows the emission of radiant energy through the hand of the therapist while they simultaneously apply the treatment. However, to date, there are no studies investigating the efficacy of applying a manual therapy protocol with a TECAR electrode in people with CNSLBP. The aim of our work is to study the efficacy of the combination of a manual therapy (MT) protocol with the simultaneous application of TECAR using a special electrode bracelet for the treatment of chronic back pain. It has been hypothesized that the simultaneous effect of the two therapeutic means through MT and high-frequency current in people with CNSLBP may further improve the therapeutic effects on pain, disability, and lumbar spine mobility than MT alone. Given the widespread use of TECAR therapy in physical therapy practice and the lack of high-level clinical studies on the subject, this research has to offer important knowledge on the subject.

## 2. Materials and Methods

### 2.1. Study Design

This was an assessor-blind three-armed randomized controlled trial (clinicaltrials.gov number: NCT05680467) conducted under the supervision of the Department of Physiotherapy of the International Hellenic University during the period December 2022–March 2023. Ethical approval was granted by the Ethics Committee of the Department of Physiotherapy of the International Hellenic University (No. EC-06/2022). Sixty participants with CNSLBP were recruited from two outpatient clinics and were randomly divided into three groups (two intervention and one control) of 20 participants each. The distribution of participants was performed with the Research Randomizer computer software (version 4) [15] by an independent researcher and thus, allocation concealment was achieved. The randomization process was conducted in small groups of three using block randomization following a 1:1 ratio in each of the three groups. None of the care providers or participants were blind to the study’s aims.

### 2.2. Participants

The inclusion criteria of the participants were (1) duration of symptoms longer than 12 weeks, (2) Numeric Pain Rating Scale (NPRS) score higher than 3 during the baseline assessment, and (3) written consent to participate in the research. The exclusion criteria of the participants were (1) neuropathic pain extending along the lower limb due to nerve root compression, (2) previous spine surgery, (3) history of spine trauma or fracture, (4) implanted pacemakers, (5) pregnancy, (6) cancer, (7) systemic musculoskeletal diseases, (8) diagnosed neurodegenerative diseases (e.g., Parkinson’s), (9) epilepsy, and (10) history of psychiatric disorders.

### 2.3. Outcome Measures

The following measurements were performed at the beginning (baseline), after two weeks, and at a one-month follow-up. Participants of both intervention groups attended a total of six sessions (three sessions per week). Each session lasted 30 min. All outcome measurements were considered primary.

#### 2.3.1. Pain with the Numeric Pain Rating Scale (NPRS)

The NPRS is an 11-point pain scale numbered from 0 to 10. The left end of the scale is marked 0 with the phrase “no pain at all”, while the right end is marked 10 with “worst possible pain”. Therefore, a higher value corresponds to more intense pain [16]. Patients marked three values on the scale, each representing pain at different time points in the last 24 h (worst pain, least pain, and pain at the time of measurement). The average of the three values was calculated and used as a variable in the research. The NPRS is widely used to measure pain in both clinical practice and research, showing high test–retest reliability and high conceptual construct validity [16].

#### 2.3.2. Functional Ability with the Roland–Morris Disability Questionnaire (RMDQ)

The disability of participants related to CNSLBP was evaluated with the Greek version of the Roland–Morris Questionnaire (RMDQ). The RMDQ consists of 24 questions related to daily activities that patients often report difficulty performing due to low back pain [17,18]. Each positive answer earns one point and the final score is calculated by adding all the points. Therefore, the higher the score, the greater the restriction. The Greek version of the questionnaire shows satisfactory reliability and validity (ICC: 0.44-0.92) [19].

#### 2.3.3. Pressure Pain Threshold (PPT) with Pressure Algometry

Pressure pain threshold (PPT) is defined as the minimal amount of pressure that produces pain. The PPT assessment was performed with a digital algometer (Wagner FPX 25 Digital Algometer; Wagner Instruments, Greenwich, CT, USA) bilaterally in the quadratus lumborum muscle, in the sacroiliac joints, and paravertebrally in the L4-L5 intervertebral space. The metal rod of the algometer was placed vertically on the site and the examiner applied gradually increasing pressure at a rate of 1 kg/s. The PPT was calculated in kg/cm^2^. Measurement was conducted according to the instructions of Imamura et al. [20,21]. The assessment of PPT in the low back region shows excellent reliability with ICC ranging from 0.86 to 0.99 [22].

#### 2.3.4. Lumbo-Pelvic Region Mobility with the Fingertip-to-Floor (FTF) Test

Changes in the lumbar spine flexion range of motion were evaluated with the Fingertip-To-Floor (FTF) test [23]. When performing the test, the examinee is asked to try to reach the ground with the fingers of their hands by leaning forward while keeping their knees and hips extended. The examiner measures with a tape the distance of the fingers from the ground. The FTF test is widely used in clinical practice to measure spinal mobility and shows high reliability and validity indices [24].

### 2.4. Experimental Protocols

#### 2.4.1. Manual Therapy (MT) Protocol

The protocol included the application of a series of MT soft tissue techniques for the lumbar area as suggested by Kaltenborn [25]. The following manual techniques were applied: lumbar soft tissue cranial and lateral mobilization with the patient in the supine position, bilateral medial lumbar soft tissue mobilization, functional massage on the quadratus lumborum muscle with the patient in a lateral position on both sides, and extension with coupled rotation and side bending mobilization in a lateral position on both sides.

#### 2.4.2. Manual Therapy with TECAR (MT + TECAR) Protocol

The participants of this group followed the same MT protocol as the first group with the addition of high-frequency current with the WinBack—TECAR device (WINBACK 3SE, Villeneuve Loubet, France). Soft tissue mobilization manipulations were applied in combination with a capacitive conventional electrode (6 cm diameter) and a special electrode bracelet that made the therapist’s own hand function as a resistive electrode. The frequency of the high-frequency current was 500 kHz, while a flexible self-adhesive ground electrode (10 cm × 18 cm) was used as a reference electrode and placed on the abdomen. The application of manual therapy protocol with TECAR is shown in Figure 1.

#### 2.4.3. Control

Participants in this group received general instructions about managing their back pain and counseling about avoiding activities that may worsen their symptoms.

### 2.5. Sample Size Determination

A total sample size of at least 45 subjects was calculated based on an a priori power analysis (G*Power 3.0.10). As a basic prerequisite for this calculation, the power (1-β) was set at 95% and the detection of a difference in the order f = 0.5 (Cohen’s f) [26]. The alpha was set at 0.05 for all tests. An additional 15% was added to the calculated sample size for the one-month follow-up after the intervention. Based on the sample size calculation mentioned above, the estimated number of participants recruited for this study was 60.

### 2.6. Statistical Analysis

Data were analyzed using SPSS Statistics for Windows, version 25.0 (SPSS Inc., Chicago, IL, USA). Normal distribution was checked using the Shapiro–Wilk test and Q-Q and P-P plots. To analyze the data, a two-way analysis of variance (ANOVA) with repeated measures was conducted. The ANOVA was applied to examine the interaction effect between the “Group” and “Time of measurement” factors. The “Group” factor was tested at three levels: manual therapy (MT) group, manual therapy plus TECAR (MT + TECAR) group, and control group. The “Time of measurement” factor was tested at three levels: baseline, second week, and one-month follow-up. If the interaction was statistically significant, the simple main effects were reported using Tukey’s post hoc test (HSD). Intention-to-treat analysis was conducted on all participants in their assigned group to ensure randomization and counter any dropout effect. In case of a dropout, previous time point values were used instead. The significance level was set at *p* < 0.05.

## 3. Results

During December 2022, a total of 78 persons were screened for eligibility. Sixty of them (76.9%) met the inclusion criteria and were randomly allocated into one of the three groups (Figure 2). No side effects from the treatments were reported in any group during the intervention. All participants attended the entirety of the intervention without any missed session visits or measurement appointments. Regarding dropouts, only one individual, who belonged to the control group, quit the program after two weeks. The demographic characteristics of participants per group are presented in Table 1.

### 3.1. NPRS Score

The statistical analysis revealed a significant interaction effect between “Group” and “Time” for the NPRS score in the two-way ANOVA analysis (*p* < 0.001). Additionally, a significant main effect was observed for the “Time” factor (*p* < 0.001) (Table 2). Tukey’s (HSD) post hoc test displayed a significant difference between groups in the NPRS score from the second week. More specifically, at the second week time point, statistically significant differences were found between the two intervention groups (“MT” vs. “MT + TECAR”) (*p* < 0.05, 95% CI) (MT + TECAR group improved further), as well as between each intervention group and the control group (“MT” vs. “control” and “MT + TECAR” vs. “control”) (*p* < 0.001, 95% CI). At the one-month follow-up time point, between-group differences were found only when comparing each intervention group with the control group (“MT” vs. “control” and “MT + TECAR” vs. “control” (*p* < 0.001, 95% CI).

### 3.2. RMDQ Score

The statistical analysis revealed a significant interaction effect between “Group” and “Time” for the RMDQ score in the two-way ANOVA analysis (*p* < 0.001). Additionally, a significant main effect was observed for the “Time” factor (*p* < 0.001) (Table 2). Tukey’s (HSD) post hoc test displayed a significant difference between groups from the second week. More specifically, at the second week time point, statistically significant differences were found between each intervention group and the control group (“MT” vs. “control” and “MT + TECAR” vs. “control”) (*p* < 0.001, 95% CI). At the one-month follow-up time point, significant differences were found between each intervention group and the control group (“MT” vs. “control” and “MT + TECAR” vs. “control”) (*p* < 0.001, 95% CI), as well as between the two intervention groups (“MT” vs. “MT + TECAR”) (MT + TECAR group improved further) (*p* < 0.05, 95% CI).

### 3.3. PPT of L4-L5 Paraspinal Intervertebral Space

The statistical analysis revealed a significant interaction effect between “Group” and “Time” for the right PPT of L4-L5 paraspinal intervertebral space in the two-way ANOVA analysis (*p* < 0.001). Additionally, a significant main effect was observed for the “Time” factor (*p* < 0.001) (Table 2). Tukey’s (HSD) post hoc test displayed a significant difference between groups from the second week. Differences in the second week were found between each intervention group and the control group (“MT” vs. “control” and “MT + TECAR” vs. “control”) (*p* < 0.05, 95% CI). At the one-month follow-up time point, significant differences were found only between the “MT + TECAR” group and the “control” group (*p* < 0.05, 95% CI). For the left PPT of L4-L5 paraspinal intervertebral space, the two-way ANOVA analysis displayed a significant “Group” x “Time” interaction effect (*p* < 0.001) as well as a main effect on the “Time” factor (*p* < 0.001) (Table 2). Tukey’s (HSD) post hoc test displayed a significant difference between groups from the second week. Differences in the second week were found between the “MT + TECAR” group and the other two groups (“MT + TECAR” vs. “MT” and “MT + TECAR” vs. “control”) (MT + TECAR group improved further) (*p* < 0.05, 95% CI). At the one-month follow-up time point, significant differences were found only between the “MT + TECAR” and the “control group” (*p* < 0.001).

### 3.4. Sacroiliac Joint PPT

The statistical analysis revealed a significant interaction effect between “Group” and “Time” for the right PPT of the sacroiliac joint in the two-way ANOVA analysis (*p* < 0.001). Additionally, a significant main effect was observed for the “Time” factor (*p* < 0.001) (Table 2). Tukey’s (HSD) post hoc test displayed a significant difference between groups from the second week. More specifically, at the second week time point, statistically significant differences were found only between the “MT + TECAR” group and the “control” group (“MT + TECAR” vs. “control”) (*p* < 0.001, 95% CI). The same differences were maintained at the one-month follow-up time point (“MT + TECAR” vs. “control”) (*p* < 0.001, 95% CI). For the left PPT of the sacroiliac joint, the two-way ANOVA analysis displayed a significant “Group” x “Time” interaction effect (*p* < 0.001) as well as a main effect on the “Time” factor (*p* < 0.001) (Table 2). Tukey’s (HSD) post hoc test displayed a significant difference between groups from the second week. More specifically, at the second week time point, statistically significant differences were found only between the “MT + TECAR” group and the “control” group (“MT + TECAR” vs. “control”) (*p* < 0.05, 95% CI). The same differences were maintained at the one-month follow-up time point (“MT + TECAR” vs. “control”) (*p* < 0.05, 95% CI).

### 3.5. Quadratus Lumborum Muscle PPT

The statistical analysis revealed a significant interaction effect between “Group” and “Time” for the right quadratus lumborum muscle PPT in the two-way ANOVA analysis (*p* < 0.001). Additionally, a significant main effect was observed for the “Time” factor (*p* < 0.001) (Table 2). Tukey’s (HSD) post hoc test displayed a significant difference between groups from the second week. More specifically, at the second week time point, statistically significant differences were found between each intervention group and the control group (“MT” vs. “control” and “MT + TECAR” vs. “control”) (*p* < 0.05, 95% CI). These differences were maintained at the one-month follow-up time point (“MT” vs. “control” and “MT + TECAR” vs. “control”) (*p* < 0.05, 95% CI). For the left quadratus lumborum muscle PPT, the two-way ANOVA analysis displayed a significant “Group” x “Time” interaction effect (*p* < 0.001) as well as a main effect on the “Time” factor (*p* < 0.001) (Table 2). Tukey’s (HSD) post hoc test displayed a significant difference between groups from the second week. More specifically, at the second week time point, statistically significant differences were found between each intervention group and the control group (“MT” vs. “control” and “MT + TECAR” vs. “control”) (*p* < 0.05, 95% CI). These differences were maintained at the one-month follow-up time point (“MT” vs. “control” and “MT + TECAR” vs. “control”) (*p* < 0.05, 95% CI).

### 3.6. FTF Test Score

The statistical analysis revealed a significant interaction effect between “Group” and “Time” for the FTF test score in the two-way ANOVA analysis (*p* < 0.001). Additionally, a significant main effect was observed for the “Time” factor (*p* < 0.001) (Table 2). The two-way ANOVA analysis displayed a significant “Group” × “Time” interaction effect *(p* < 0.001) for the FTF test score, while a main effect on the “Time” factor was observed (*p* < 0.001) (Table 2). Tukey’s (HSD) post hoc test displayed a significant difference between groups in the FTF test score from the second week. More specifically, at the second week time point, statistically significant differences were found between the two intervention groups (“MT” vs. “MT + TECAR”) (MT + TECAR group improved further), as well as between each intervention group and the control group (“MT” vs. “control” and “MT + TECAR” vs. “control”) (*p* < 0.05, 95% CI). At the one-month follow-up time point, significant differences were found between the two intervention groups (“MT” vs. “MT + TECAR”) (*p* < 0.05, 95% CI), as well as between “MT + TECAR” and control groups (“MT + TECAR” vs. “control”) (*p* < 0.001, 95% CI).

## 4. Discussion

In our research, the same MT protocol was applied with and without the application of a TECAR high-frequency current to investigate whether the simultaneous application of the two methods could be more effective, compared to the individual application of MT, in the treatment of symptoms in people with CNSLBP. Modern TECAR devices are widely accepted in clinical practice, allowing the simultaneous application of high-frequency current with special electrodes, along with therapeutic applications, such as manipulation, mobilization, or therapeutic exercise. However, their efficacy has not been proven in clinical studies. Research data surrounding the applications of TECAR both in the treatment of CNSLBP [12,14,27] and its use in other musculoskeletal disorders [10] concern the conventional applications of these devices using simple capacitive and resistive electrodes.

The mechanisms to achieve analgesia are known. In manual therapy, the therapeutic manipulations are able, through the mobilization of the tissues and stimulation of the sensory receptors, to block the noxious stimuli of the pain receptors and simultaneously increase the local circulation, the temperature, and the local metabolism at the cellular level [28,29]. However, the analgesic effect of manual therapy is mainly based on mechanical stimuli, while the thermal effect produced has been shown to be negligible and limited only to the superficial muscles [28].

On the other hand, the high-frequency current emitted by a TECAR device produces an intense thermal effect at different tissue levels (muscles, tendons, cartilage, joints, and bones). Generating this deep heat through radio frequency emission has been shown to greatly increase cellular metabolism and provide intense analgesic effects and healing action, which leads to reduced recovery time [10].

The study was conducted on the assumption that the simultaneous emission of the high-frequency current with another means such as the therapeutic manipulations of the care provider combines the therapeutic effects of manual therapy with the effect of deep heat and possibly enhances the analgesic effect. By maintaining consistent conditions throughout the application of the ΜΤ protocol, both with and without the application of TECAR therapy, and at the same time by ensuring sample homogeneity in terms of initial measurements and demographic characteristics, any distinction between the groups can be attributed to the cumulative impact of the MT and TECAR therapeutic means.

The results of this research show that the application of the same MT protocol using a conventional capacitive and a resistive electrode bracelet further improved the pain of participants. The NPRS score improved significantly in both intervention groups compared to the control group in the second week, while this difference was maintained at the one-month follow-up (Table 2). However, the decrease in NPRS score noted in the MT + TECAR group (−2.85 points corresponding to a reduction of 49.23%) was greater than that of the MT group (−2 points, 32.78% reduction) with a statistically significant difference in the second week. A possible explanation for this is that the combined effect of MT and deep heat produced by TECAR led to the maximum analgesic effect further reducing the NPRS score. The difference between the two intervention groups in the second week is clinically important. According to Ostelo et al. [30], the minimally clinically important change in the NPRS score in individuals with CNSLBP is 2.5 points, which was observed only in the MT + TECAR group. However, it is worth mentioning that the difference in the NPRS score between the two groups did not remain statistically significant at the one-month follow-up, possibly due to the short intervention period. Perhaps, the two weeks were not enough to maintain this difference at the same levels one month after the intervention. We concluded therefore that more long-term interventions should be implemented in the future to capture the effects of this treatment protocol in people with CNSLBP.

Our results agree with those of Tashiro et al. [14], who also found similar positive effects on pain by adding TECAR to a therapeutic exercise program in individuals with CNSLBP. However, they used conventional TECAR electrodes without the simultaneous application of other therapeutic means through a special electrode bracelet as in our study, so a direct comparison of the results is not possible.

Corresponding differences between the groups were also noted in the RMDQ score. Both intervention groups showed statistically significant differences compared to the control group from the second week. These differences remained statistically significant during the one-month follow-up measurement. A difference was also found from the second week between the two intervention groups with the MT + TECAR group showing an improvement in the score compared to the initial measurement by 51.91% (6.1 points decrease) compared to the MT group which improved by 36.4% (4.15 points decrease). However, the differences between the two intervention groups appeared statistically significant only at the one-month follow-up with bigger mean differences between groups at the one-month follow-up related to the second week (3.05 and 2.6, respectively). This fact can be partially explained by the generally lower levels of pain experienced after the intervention by the participants of the MT + TECAR group compared to the participants of the MT group, which is also reflected by the results of the NPRS score. The difference between the two intervention groups at the one-month follow-up is also clinically significant. According to Ostelo et al. [30], the minimally clinically important change in the RMDQ score in individuals with CNSLBP is 3.5 points. This difference was only noted in the participants of the MT + TECAR group and not in the participants of the MT group (Table 2). The results of our research agree with those of other researchers who also found a corresponding reduction in disability with TECAR therapy in combination with exercise [14].

Regarding the PPT values of the L4-L5 paraspinal intervertebral space on the right side, statistically significant differences were detected between the two intervention groups and the control group in the second week, while at the one-month follow-up, differences were detected only between the MT + TECAR group and the control group. A possible explanation is the greater increase in the PPT value noted in the MT + TECAR group compared to the other two groups. It appears that even if the difference between the two intervention groups in the second week did not appear statistically significant, the improvement in the MT + TECAR group was maintained at high levels one month later in contrast to the MT group. This possibly explains why the MT group did not show statistically significant differences compared to the control group at the one-month follow-up (Table 2). Therefore, it appears that the combined effect of MT and TECAR made the improvements in PPT last longer.

Concerning the L4-L5 paraspinal intervertebral space PPT on the left side, in the second week, differences were detected between the two intervention groups as well as between the MT + TECAR group and the control group, which remained statistically significant during the one-month follow-up (Table 2). Consequently, the PPT value in the MT + TECAR group (presented numerically in Table 2) was much higher than the other two groups, and this improvement was maintained even after one month following the intervention. On the other hand, the improvement in value noted in the MT group in the second week was not maintained one month later, which is confirmed by the fact that there were no differences between the MT group and the control group at the one-month follow-up.

Regarding the sacroiliac joint PPT on both the right and left sides, the only differences noted in the second week were between the MT + TECAR group and the control group. In fact, these differences remained statistically significant during the one-month follow-up, which means that the improvement noted in the participants of the MT + TECAR group (shown numerically in Table 2) was greater than that noted in the participants of the other two groups. Even if no statistically significant differences were noted between the two intervention groups at any time point, the fact that no corresponding differences were noted between the MT group and the control group is for us an indirect indication that the MT + TECAR approach appeared more effective than the other two in increasing PPT.

The improvements observed in the PPT values in both the L4-L5 paraspinal intervertebral space and the sacroiliac joint in the participants of the MT + TECAR group appear to be due to the combined effect of MT and TECAR. It is possible that the combination of mechanical (MT) and thermal (TECAR) effects at the same time increased the PPT values more effectively and for a longer period.

Regarding the quadratus lumborum muscle PPT, both on the right and left sides, differences between the groups were detected in the second week between the two intervention groups and the control group, while these differences remained statistically significant at the one-month follow-up. However, although Table 2 shows that the improvement of the quadratus lumborum muscle PPT was greater in the participants of the MT + TECAR group compared to those of the MT group, this difference did not appear to be statistically significant. Therefore, we can say that the additional application of TECAR did not seem to add a significant additional positive effect to the improvement of the quadratus lumborum muscle PPT. On the question of why the addition of TECAR seemed to contribute more to the improvement of PPT values in the L4-L5 paraspinal intervertebral space as well as the sacroiliac joint while the same did not happen with the PPT of the quadratus lumborum muscle, a possible answer is as follows: The first two points refer to areas of the body that also contain non-contractile elements (tendons, joints, and bones), while the quadratus lumborum muscle contains soft tissues. The special bracelet electrode used to apply most of the MT manipulations in the MT + TECAR protocol was a resistive electrode, which according to the manufacturer causes a temperature increase more in areas of the body poor in fluid, such as ligaments, joints, and bones, and less so in muscles [12]. However, this interpretation remains to be proven as there are recent data, according to which the resistive mode may increase the intramuscular blood flow in healthy adults more than the capacitive mode [31]. More research in the future can study the effects on the different tissues of the combined MT and TECAR protocols using special electrodes depending on the target areas and on the operating method of the device (capacitive or resistive). In a recent study, no differences were found between the two operating methods of TECAR in individuals with CNSLBP [12]. However, the researchers did not study the combined effect of MT and TECAR and applied their protocol with conventional TECAR electrodes. We believe that more research is needed on different combined MT and TECAR protocols using specific capacitive and resistive electrodes to have a clearer picture of their analgesic effect on different tissues.

Finally, the results of the FTF test score showed that it improved significantly in both intervention groups compared to the control group in the second week, while this difference was maintained at the one-month follow-up (Table 2). However, the improvement in the FTF test score found in the MT + TECAR group was greater than that of the MT group with a statistically significant difference in the second week (−9.25 cm, 58.91% reduction) compared to the corresponding reduction observed in the participants of the MT group (−4.3 cm, 27.21% reduction). This further improvement in lumbar mobility observed in participants of the MT + TECAR group may be due to the thermal effect of TECAR. It is known that the application of TECAR increases joint mobility, especially in stiff joints, and also the range of motion [10]. However, it is not clear from our own research whether this improvement is due to the combined effect of MT and TECAR or the effect of TECAR alone (as for this, there would have to be another group receiving TECAR treatment alone). Finally, it is worth noting that at the one-month follow-up, differences were only found between the MT + TECAR group and the other two groups, which means that the individuals in this group not only showed greater improvement compared to the individuals in the other two groups, but this improvement also had a longer duration.

This study had several limitations. The fact that neither participants nor care providers were blinded to the aims of the study was a threat. The small number of participants in combination with a short follow-up period was also a significant limitation of this research. Another limitation concerns the selection of the FTF test. This test is affected by a number of factors such as hamstring tightness or sacroiliac joint dysfunction, while it is limited to measure only the trunk bending movement without taking into account the rest of the movements. However, the FTF test was chosen because of its high reliability in people with low back pain (ICC = 0.99), as well as its ease of use [23].

## 5. Conclusions

The application of an MT protocol with TECAR using a resistive special bracelet electrode seems to improve pain and disability further than conventional MT in individuals with CNSLBP. The positive effect seems to be due to the combination of mechanical and thermal effects offered simultaneously by the two therapeutic methods. Moreover, the MT + TECAR combination seems to improve local sensitivity in the lumbar region more effectively. However, the greatest effect was found in fluid-poor tissue areas (such as tendons, joints, and bones) and less so in muscles. Finally, the thermal effect of TECAR seems to be responsible for improving mobility in the lumbo-pelvic region. In most measures, the improved values between the participants who followed the MT protocol with TECAR and the other groups were maintained for one month after the end of the treatments. However, the short duration of the intervention only gives us some indications without allowing safe conclusions to be drawn. More long-term research in the future might be able to shed more light on the matter.

## Figures and Tables

**Figure 1 medicina-59-01275-f001:**
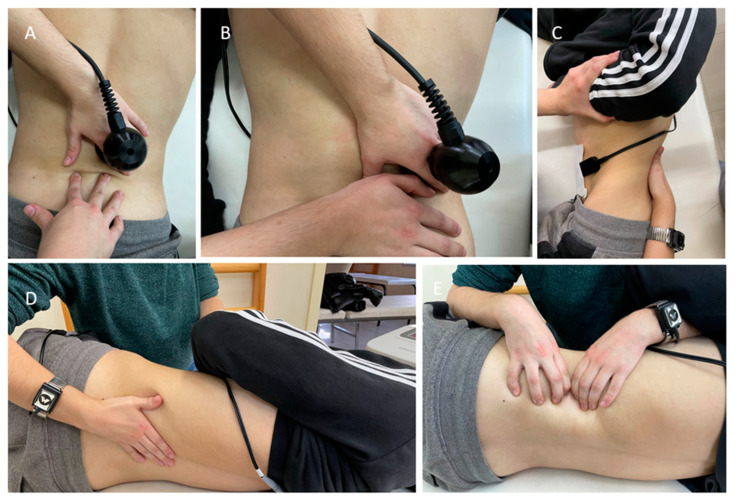
MT protocol with TECAR: (**A**) lumbar cranial and lateral mobilization through capacitive conventional electrode, (**B**) medial lumbar soft tissue mobilization with the capacitive conventional electrode, (**C**,**D**) extension with coupled rotation and side bending mobilization through resistive bracelet electrode, and (**E**) functional massage on the quadratus lumborum muscle through resistive bracelet electrode.

**Figure 2 medicina-59-01275-f002:**
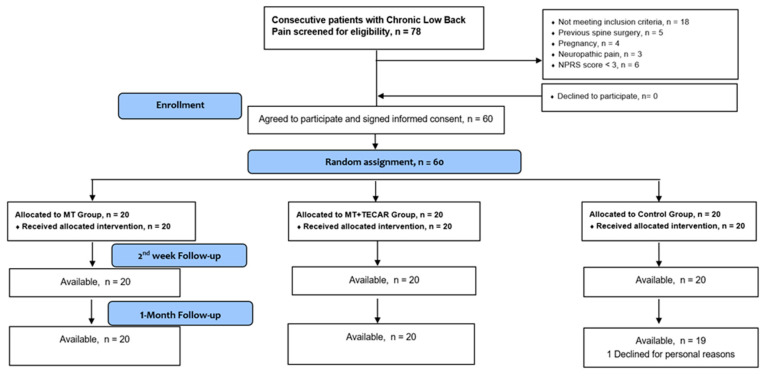
CONSORT flow diagram of the study.

**Table 1 medicina-59-01275-t001:** Demographic characteristics of the three groups.

Demographics	Group 1 (MT)	Group 2 (MT + TECAR))	Group 3 (Control)
Age (years)	37.85 (2.62)	39.20 (2.63)	38.10 (2.36)
Sex (Men/Women)	35% (*n* = 7) Men	30% (*n* = 6) Men	35% (*n* = 7) Men
	65% (*n* = 13) Woman	70% (*n* = 14) Woman	65% (*n* = 13) Woman
Affected side (Right/Left/Both)	60% (*n* = 12) Right	70% (*n* = 14) Right	75% (*n* = 15) Right
	25% (*n* = 5) Left	20% (*n* = 4) Left	20% (*n* = 4) Left
	15% (*n* = 3) Both	10% (*n* = 2) Both	5% (*n* = 1) Both
Symptoms Duration (months) (3–6/6–12/More than 12)	30% (*n* = 6) 3–6	35% (*n* = 7) 3–6	40% (*n* = 8) 3–6
	50% (*n* = 10) 6–12	40% (*n* = 8) 6–12	40% (*n* = 8) 6–12
	20% (*n* = 4) More than 12	25% (*n* = 5) More than 12	20% (*n* = 4) More than 12
Previous Physiotherapy (Yes/No)	80% (*n* = 16) Yes	85% (*n* = 17) Yes	85% (*n* = 17) Yes
	20% (*n* = 4) No	15% (*n* = 3) No	15% (*n* = 3) No

MT = manual therapy; MT + TECAR = manual therapy plus TECAR.

**Table 2 medicina-59-01275-t002:** The mean (SD) values of the outcome measures for each time point for the three groups. The “Group” × “Time” interaction *p*-Values (95% CI), as well as between-group pairwise comparisons *p*-Values, for each time point are also presented.

	Baseline	Week 2	1-Month Follow-Up
NPRS score (SD)			
Group 1 (MT)	6.10 (1.33)	4.10 (1.11)	3.95 (0.99)
Group 2 (MT + TECAR)	6.05 (1.39)	3.20 (1.24)	3.30 (1.08)
Group 3 (Control)	5.95 (1.39)	5.50 (1.14)	6.25 (1.02)
*p*-Value (Interaction)	<0.001		
	0.02		
	0.00		
*p*-Value (between groups)	*p* ^a,b,c^ > 0.05	*p* ^a,b,c.^*	*p* ^b,c.^ *
RMDQ score (SD)			
Group 1 (MT)	11.40 (1.42)	7.25 (2.69)	8.80 (2.96)
Group 2 (MT + TECAR)	11.75 (1.94)	5.65 (2.90)	6.10 (2.90)
Group 3 (Control)	11.80 (1.67)	10.95 (1.82)	12.65 (1.81)
*p*-Value (Interaction)	<0.001		
*p*-Value (between groups)	*p* ^a,b,c^ > 0.05	*p* ^b,c.^ *	*p* ^a,b,c.^ *
L4-L5 paraspinal intervertebral space PPT, kg/cm^2^ Right			
Group 1 (MT)	4.10 (1.57)	5.64 (1.76)	5.14 (1.57)
Group 2 (MT + TECAR)	3.99 (1.24)	6.03 (1.31)	6.07 (1.28)
Group 3 (Control)	4.06 (1.27)	4.38 (1.31)	4.33 (1.27)
*p*-Value (Interaction)	<0.001		
	0.00		
	0.00		
*p*-Value (between groups)	*p* ^a,b,c^ > 0.05	*p* ^b,c.^*	*p* ^c^ *
L4-L5 paraspinal intervertebral space PPT, kg/cm^2^ Left			
Group 1 (MT)	4.50 (1.45)	5.34 (1.51)	5.10 (1.53)
Group 2 (MT + TECAR)	4.74 (0.86)	6.38 (0.69)	6.28 (0.69)
Group 3 (Control)	4.48 (1.23)	4.72 (1.21)	4.25 (0.98)
*p*-Value (Interaction)	<0.001		
*p*-Value (between groups)	*p* ^a,b,c^ > 0.05	*p* ^a,c.^*	*p* ^a,c.^*
Sacroiliac joint PPT, kg/cm^2^ Right			
Group 1 (MT)	3.83 (1.05)	4.82 (1.07)	4.57 (1.07)
Group 2 (MT + TECAR)	3.93 (1.11)	5.46 (1.10)	5.36 (1.13)
Group 3 (Control)	3.98 (0.89)	4.10 (0.90)	4.00 (0.91)
*p*-Value (Interaction)	<0.001		
*p*-Value (between groups)	*p* ^a,b,c^ > 0.05	*p* ^c.^ *	*p* ^c.^ *
Sacroiliac joint PPT, kg/cm^2^ Left			
Sacroiliac joint PPT, kg/cm2 Right			
Group 1 (MT)	4.23 (1.31)	5.19 (1.26)	4.94 (1.17)
Group 2 (MT + TECAR)	4.31 (1.10)	5.63 (1.18)	5.44 (1.19)
Group 3 (Control)	4.23 (1.01)	4.41 (1.06)	4.35 (1.01)
*p*-Value (Interaction)	<0.001		
	*p*-Value (Interaction)		
	<0.001		
*p*-Value (between groups)	*p* ^a,b,c^ > 0.05	*p* ^c.^ *	*p* ^c.^ *
Quadratus lumborum muscle PPT, kg/cm^2^ Right			
Group 1 (MT)	4.05 (0.89)	4.86 (0.98)	4.76 (0.94)
Group 2 (MT + TECAR)	3.83 (1.21)	5.36 (1.16)	5.29 (1.13)
Group 3 (Control)	3.76 (1.00)	4.06 (0.86)	3.97 (0.85)
*p*-Value (Interaction)	< 0.001		
	*p*-Value (Interaction)		
	< 0.001		
*p*-Value (between groups)	*p* ^a,b,c^ > 0.05	*p* ^b,c.^*	*p* ^b,c.^*
Quadratus lumborum muscle PPT, kg/cm^2^ Left			
Quadratus lumborum muscle PPT, kg/cm^2^ Right			
Quadratus lumborum muscle PPT, kg/cm^2^ Right			
Quadratus lumborum muscle PPT, kg/cm^2^ Right			
Group 1 (MT)	4.27 (1.04)	5.12 (0.96)	4.98 (0.97)
Group 2 (MT + TECAR)	4.09 (0.92)	5.67 (0.97)	5.52 (0.93)
Group 3 (Control)	4.00 (0.94)	4.19 (0.89)	4.04 (0.84)
*p*-Value (Interaction)	<0.001		
	*p*-Value (Interaction)		
	*p*-Value (Interaction)		
*p*-Value (between groups)	*p* ^a,b,c^ > 0.05	*p* ^b,c.^*	*p* ^b,c.^*
FTF test score			
Group 1 (MT)	15.80 (4.39)	11.50 (4.05)	13.50 (4.09)
Group 2 (MT + TECAR)	15.70 (2.13)	6.45 (1.50)	7.15 (1.89)
Group 3 (Control)	15.50 (2.62))	14.50 (2.80)	15.35 (4.00)
*p*-Value (Interaction)	>0.001		
*p*-Value (between groups)	*p* ^a,b,c^ > 0.05	*p* ^a,b,c.^*	*p* ^a,c.^ *

Between-group comparison: a = Group 1 vs. Group 2; b = Group 1 vs. Group 3; c = Group 2 vs. Group 3. * Between-group significant comparisons in the post hoc testing.

## Data Availability

The datasets generated and analyzed during the current study are available from the corresponding author upon reasonable request.

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
