# Peer review of "Short-Term Effects of Manual Therapy plus Capacitive and Resistive Electric Transfer Therapy in Individuals with Chronic Non-Specific Low Back Pain: A Randomized Clinical Trial Study"

_medicina, 2023, doi:10.3390/medicina59071275_

Round 1

Reviewer 1 Report

After reviewing the paper it is necessary to correct the following points:

Line 50: reference 1 may not be correct, it may be reference 2.

Line 51: reference 2 may not be correct, it may be reference 1.

Line 212: it is not specified which is better (MT or MT+TECAR).

Line 225: it is not specified who is better (MT or MT+TECAR).
Line 240: not specified in favor of who is better (MT or MT+TECAR)
Line 280: it is not specified in favor of who is better (MT or MT+TECAR).

Suggestions for explanations:

Line 376-384): it is possible that the small number of patients causes this difference in results (on the left side there are 5-4-4 patients in each group and on the right side 12-14-15).
Line 408 and following: another explanation would be that tendon, joint and bone causes are very frequently responsible for low back pain and the action of electrical stimulation therapy on these structures is effective in relieving it.
Line 426 and following: the same explanation as above can be applied.

Add information to the table:
Table 2: the statistical significance in the P-value (between groups) is not specified, according to the text some have a value p<0.05 and others p<0.001.

Bibliography: according to publication standards of the journal, 2 authors are cited:
Correct citations that have more than 2 authors.
Line 500: space between 2017 and 30 (2017,30 for 2017, 30).
Line 511: delete (:) between 1375, :39
Line 540: space between 91 and 1243 (91,1243 by 91, 1243).
Line 554: space between 2005 and 19 (2005,19 by 2005, 19).

Author Response

Reviewer #1comments

Thank you for your feedback regarding our article. We made the necessary corrections to the text and resubmitted the revised manuscript. We responded to all your comments. Changes in text have been highlighted, so that you may track them. Below are our answers to your comments.

Line 50: reference 1 may not be correct, it may be reference 2.

Thank you for your comment. We reworded the first sentence of the introduction and changed the reference [1] so that it exactly matches the description of the definition of Chronic Non-Specific Low Back Pain.

Line 51: reference 2 may not be correct, it may be reference 1.

Reference 2 is correct.

Line 212: it is not specified which is better (MT or MT+TECAR).

Thank you for pointing out, edited according to your suggestions. We added a clarifying sentence (line numbers, 215-216).

Line 225: it is not specified who is better (MT or MT+TECAR).

Thank you for pointing out, edited according to your suggestions. We added a clarifying sentence (line numbers, 229-230).

Line 240: not specified in favor of who is better (MT or MT+TECAR)

Edited according to your suggestions. We added a clarifying sentence (line number, 244).

Line 280: it is not specified in favor of who is better (MT or MT+TECAR).

Edited according to your suggestions. We added a clarifying sentence (line numbers, 286-287).

Suggestions for explanations:

Line 376-384): it is possible that the small number of patients causes this difference in results (on the left side there are 5-4-4 patients in each group and on the right side 12-14-15).

Thank you for your comment. Participants were instructed at the start of the study to indicate on which side they experienced the most severe symptoms. People who reported a specific affected side (left or right) did not rule out experiencing symptoms on both sides of the body. Only those who could not identify the affected side and felt their pain equally on both sides reported “both sides”. Therefore, even if most of the participants (as indicated in Table 2) mentioned the right side as their affected side, it does not mean that they did not experience symptoms (even if less intense) on the left side as well. Especially in the case of chronic pain it is extremely unlikely that the symptoms will be localized to only one side of the body. With this logic we treated the symptoms of each side as separate variables and interpreted the results in the same way based on the findings of the statistical analysis. Another reason we adopted this logic is because we would have to question all the findings that involved the PPT variables on the left side.

Line 408 and following: another explanation would be that tendon, joint and bone causes are very frequently responsible for low back pain and the action of electrical stimulation therapy on these structures is effective in relieving it.

Electrical stimulation therapy is a term that is more commonly used in the application of low-frequency currents that aim to relieve through the stimulation of a muscle without, however, a strong thermal effect. TECAR, on the other hand, belongs to the high-frequency currents and its effect on the tissues is not stimulating but is mainly based on the production of heat in depth (through the emission of radio waves the electrical energy is converted into heat inside the body achieving an intense thermal effect in depth). In relation to the diathermy devices that are the ancestor of TECAR, the TECAR technology (according to the instructions of the manufacturers of devices of various companies) through capacitive and resistive electrical transfer modes enables the user to heat more the tissues rich in fluid (e.g., muscles) or weak (bones and tendons) by changing the function of the device and the corresponding electrode from capacitive to resistive. Therefore, there is a theoretical background behind the justification we have given in this paragraph which for us is important to point out. We also believe that this interpretation adds value to our article as even though the manufacturers of the TECAR devices suggest the application of different types of radio wave emission for different target tissues, this has so far not been confirmed by research. Other studies that have applied TECAR protocols for low back pain report that both modes act in the same way on the tissues with no differences between them (Barassi et al. 2020).

Line 426 and following: the same explanation as above can be applied.

For the same reasons we mention in the answer to your previous comment we believe that it was the production of the deep heat that allowed the improvement in the mobility of the lumbosacral area. For this reason we left the text as it is.

Add information to the table: Table 2: the statistical significance in the P-value (between groups) is not specified, according to the text some have a value p<0.05 and others p<0.001.

Due to the design of our study (three different groups with three different measurement times), we cannot include all p-values in the text of the results or in Table 2 because the volume of data is too large. Each possible combination between the different groups at each measurement time also has a separate p-value that is the result of the post-hoc tests. The way the results of the post-hoc tests are shown in Table 2 (Between group comparison: a= Group 1 VS Group 2; b=Group 1VS Group 3; c=Group 2 VS Group 3. * Between groups significant comparisons in the post-hoc testing), is equivalent to that found in another research to make the table easy to read (Lytras et al. 2023). However, if you insist, we can add an additional table (Table 3) or an appendix to the article which will contain the mean differences, the p-values, and the 95% CIs for multiple comparisons between groups in each time point for all variables.

Bibliography: according to publication standards of the journal, 2 authors are cited: Correct citations that have more than 2 authors.

Thank you for your comment. We followed the author’s instructions section of the journal and used MDPI Chicago style on Zotero. According to this style, all the authors should be cited in the reference list. Additionally, we have checked other published articles in Medicina Journal, and all of them cite more than two authors in the reference list (https://www.mdpi.com/1648-9144/59/7/1238). If this is not correct, please provide us with clear instructions to correct the reference list accordingly.

Line 500: space between 2017 and 30 (2017,30 for 2017, 30).

Edited according to your suggestions (line number 516).

Line 511: delete (:) between 1375, :39

Edited according to your suggestions (line number 527).

Line 540: space between 91 and 1243 (91,1243 by 91, 1243).

Edited according to your suggestions (line number 556).

Line 554: space between 2005 and 19 (2005,19 by 2005, 19).

Edited according to your suggestions (line number 570).

References

Barassi G, Mariani C, Supplizi M, Prosperi L, Di Simone E, Marinucci C, Pellegrino R, Guglielmi V, Younes A, Di Iorio A. Capacitive and Resistive Electric Transfer Therapy: A Comparison of Operating Methods in Non-specific Chronic Low Back Pain. Adv Exp Med Biol. 2022;1375:39-46. doi: 10.1007/5584_2021_692. PMID: 35147930.

Lytras D, Sykaras E, Iakovidis P, Kasimis K, Kottaras A, Mouratidou C. Comparison of two different manual techniques for an exercise program for the management of chronic neck pain: A randomized clinical trial study. J Back Musculoskelet Rehabil. 2023;36(1):199-216. doi: 10.3233/BMR-220003. PMID: 36120764.

Reviewer 2 Report

The manuscript entitled: “Short-term Effects of Manual Therapy plus Capacitive and Resistive Electric Transfer Therapy in Individuals with Chronic Non-specific Low Back Pain: A Randomized Clinical Trial Study “has an interesting subject and provides valuable findings to readers. First of all, I think this topic is appropriate for orthopedic section of the journal rather than neurology. However, there are questions and points which should be considered as follows.

Abstract:

The heading Background and Objectives should be merged with Aim heading. It is also necessary to briefly explain the role of TECAR in the CNSLBP or why it is used.

Introduction:

Line 63, Move the word “Therapy” after the abbr. of TECAR.

Methods:

Outcome measures: line 110: you have mentioned that each therapy session lasted 30 minutes. However, you should mention the duration of tecar therapy. In most studies it has been used for 8-10 minutes. Did you apply it for 30 minutes? 

2.3.1. Line 116, did you assess the patients’ pain at rest or asked them about the pain perception during their activities, or asked about pain perception as a whole? Please state that. Also, mention the area where pain is reported.

2.3.2 Line 136: It seems that it should be 2.3.4 not 2.3.2. Also, please explain why only the FTF was chosen to assess lumbo-pelvic mobility? This test is influenced by a number of factors such as hamstrings tightness. How did you control that? On the other hand it assesses only lumbar flexion not the other movements such as lumbar rotations or lateral bending or extension; or even movements in SI joint as one of your targeted area of treatment(FABER test is more appropriate for it). Please, identify these limitations or explain why you selected the FTF test.

Experimental protocols

Line 154-158: Identify the active and reference electrode sizes. Did you apply tecar with different frequencies? In most studies it is applied with a fixed frequency for example 448 kHz or 500 kHz, etc.

Discussion

Line 322: you have stated: “this research show that the application of the same MT protocol using a resistive electrode bracelet further improved the pain of participants. How did you conclude it? It seems that you have used both capacitive and resistive electrodes, and the capacitive electrodes have also been used for parasinal muscles (Fig. 1 A and B). Indeed, using capacitive electrodes are more suitable for tissues like muscles, but why you have concluded NPRS data based on applying different electrodes?

Line 415-418: you can also benefit from the results of the below article in 2020 which sated that more intramuscular blood flow is produced when using the resistive electrode.

Clijsen R, Leoni D, Schneebeli A, Cescon C, Soldini E, Li L, Barbero M. Does the Application of Tecar Therapy Affect Temperature and Perfusion of Skin and Muscle Microcirculation? A Pilot Feasibility Study on Healthy Subjects. J Altern Complement Med. 2020 Feb;26(2):147-153.

Author Response

Reviewer #2 comments

Thank you for your feedback regarding our article. We made the necessary corrections to the text and resubmitted the revised manuscript. We responded to all your comments. Changes in text have been highlighted, so that you may track them. Below are our answers to your comments.

Abstract:

The heading Background and Objectives should be merged with Aim heading. It is also necessary to briefly explain the role of TECAR in the CNSLBP or why it is used.

Edited according to your suggestions (line numbers: 15-19).

Introduction:

Line 63, Move the word “Therapy” after the abbr. of TECAR.

Edited according to your suggestions (line number, 64).

Methods:

Outcome measures: line 110: you have mentioned that each therapy session lasted 30 minutes. However, you should mention the duration of tecar therapy. In most studies it has been used for 8-10 minutes. Did you apply it for 30 minutes?

Thank you for your comment. You are correct in stating that 30 minutes is a long time to implement TECAR. In the research we found, TECAR application protocols in people with CNSLBP ranged from 15 to 20 minutes and included the application of both operation modes (capacitive – resistive) with the following dosage: 5 min capacitive and 10 min resistive (Wachi et al. 2022), or 10 min capacitive and 10 min resistive (Barassi et al. 2022; Notarnicola et al. 2017). We followed the second option, i.e., the net TECAR application time was 20 minutes. However, all the above protocols of the other researchers were performed with conventional electrodes and with the patient in a prone position with the metal grounding electrode placed on the abdomen while there were no position changes during the treatment. In our protocol, on the other hand, adhesive grounding was used, which takes a few minutes to place (if we consider the local cleaning and disinfection of the skin before placing the adhesive electrode). Additionally, the frequent changes of the patient's position on the bed for the application of the protocol added extra minutes to each treatment session (often during the change of positions when losing the electrode contact with the body the timer stopped for a few seconds). Therefore, wanting to apply a TECAR therapy net time of 20 minutes (10 minutes capacitive and 10 minutes resistive), we added an additional 10 minutes calculating the additional time required for the above procedures. With this in mind, we reported that the total application time of the TECAR protocol was 30 minutes. If you consider this information to be misleading to the readers of the journal, we can correct it. However, the 30 minutes correspond to the actual time of each session.

2.3.1. Line 116, did you assess the patients’ pain at rest or asked them about the pain perception during their activities, or asked about pain perception as a whole? Please state that. Also, mention the area where pain is reported.

Thank you for your comment. We assessed the intensity of low back pain in the last 24 hours without asking for further clarification during the measurement about the characteristics of this pain or the factors (activities) that aggravate the symptoms (specific activities etc.). Use of the NPRS was performed following the guidelines set by Childs et al. (2005). We asked the patients to fill in for the last 24 hours three values, one for the most intense pain, one for the lowest, and one that related to the intensity of their pain at the time of the measurement. The average of these three values was the measurement variable. Since this information was not mentioned in the original text, we added a clarifying sentence to the description of the tools (line numbers: 117-120).

2.3.2 Line 136: It seems that it should be 2.3.4 not 2.3.2. Also, please explain why only the FTF was chosen to assess lumbo-pelvic mobility? This test is influenced by a number of factors such as hamstrings tightness. How did you control that? On the other hand it assesses only lumbar flexion not the other movements such as lumbar rotations or lateral bending or extension; or even movements in SI joint as one of your targeted area of treatment (FABER test is more appropriate for it). Please, identify these limitations or explain why you selected the FTF test.

You are correct about “2.3.2”, it was changed to “2.3.4” (line number, 139). Regarding the choice of the FTF test, below we explain the logic by which it was chosen: The FTR assesses the mobility of both the spine and the pelvis in the forward bending movement of the trunk. Based on our clinical experience, people with chronic low back pain show reduced mobility in the area due to the prolonged duration of the symptoms, which is related not only to the reduced mobility of the lumbar spine but also to the mobility of the pelvis. Given the ease of use and the excellent reliability of this test (ICC = 0.99) in people with low back pain (Perret et al. 2001), we considered that it was suitable for our research. However, we agree that the test you suggest may be more appropriate but since we cannot change the tool used retrospectively and following the prompt in your comment we have added a relevant phrase to the limitation section (see line numbers: 457-461).

Experimental protocols

Line 154-158: Identify the active and reference electrode sizes. Did you apply tecar with different frequencies? In most studies it is applied with a fixed frequency for example 448 kHz or 500 kHz, etc.

Below we provide the information you request. Regarding the size of the electrodes, the sizes are as follows: The conventional capacitive electrode had a diameter of 6cm. The adhesive flexible ground had dimensions of 10 X 18cm (the dimensions were added to the text (see line numbers 158 and 161). The special electrode bracelet does not have a specific dimension because it turns the entire hand of the therapist into a resistive electrode, the dimension of which fluctuates every moment depending on the contact surface of the hand with the patient's body. Since the arm wearing the bracelet was not static but was performing therapeutic manipulations simultaneously with the TECAR emission, the “dimension” of the electrode was variable and therefore cannot be determined.

You are right about the frequencies. The TECAR protocol was implemented in fixed frequency at 500khz (corrected in text: line number 160). The frequencies we mentioned in the original text (300khz and 1Mhz) were related to other applications of the device that are not related to the specific protocol.

Discussion

Line 322: you have stated: “this research show that the application of the same MT protocol using a resistive electrode bracelet further improved the pain of participants.“  How did you conclude it? It seems that you have used both capacitive and resistive electrodes, and the capacitive electrodes have also been used for parasinal muscles (Fig. 1 A and B). Indeed, using capacitive electrodes are more suitable for tissues like muscles, but why you have concluded NPRS data based on applying different electrodes?

Thank you for your comment. We corrected that sentence by adding the conventional capacitive electrode (see line number: 332. As we mention at another point in the discussion (line numbers: 423-425) most of the MT protocol (i.e., most of the manipulations) was applied with the bracelet electrode (the capacitive electrode was mainly used as a preparation). In any case, the important thing for us was the logic of the application of the protocol that we applied, which was based on our attempt to integrate the application of TECAR into the program of MT and not as a separate part as is done in most studies. Even the use of the conventional capacitive electrode was done in such a way as to mimic the protocol of the MT without TECAR (as shown in Figure 1 the head of the device was placed between the index and the middle fingers so that the surface of the electrode was added to the surface of the palm applying the technique).

Line 415-418: you can also benefit from the results of the article of Clijsen et al. (2020) which sated that more intramuscular blood flow is produced when using the resistive electrode.

Thank you for your comment. We added a relevant sentence to the discussion in the article and included the suggested reference (line numbers: 427-429 and 572-574).

References

Barassi G, Mariani C, Supplizi M, Prosperi L, Di Simone E, Marinucci C, Pellegrino R, Guglielmi V, Younes A, Di Iorio A. Capacitive and Resistive Electric Transfer Therapy: A Comparison of Operating Methods in Non-specific Chronic Low Back Pain. Adv Exp Med Biol. 2022;1375:39-46. doi: 10.1007/5584_2021_692. PMID: 35147930.

Childs JD, Piva SR, Fritz JM. Responsiveness of the numeric pain rating scale in patients with low back pain. Spine (Phila Pa 1976). 2005 Jun 1;30(11):1331-4. doi: 10.1097/01.brs.0000164099.92112.29. PMID: 15928561.

Notarnicola A, Maccagnano G, Gallone MF, Covelli I, Tafuri S, Moretti B. Short term efficacy of capacitive-resistive diathermy therapy in patients with low back pain: a prospective randomized controlled trial. J Biol Regul Homeost Agents. 2017 Apr-Jun;31(2):509-515. PMID: 28685560.

Perret C, Poiraudeau S, Fermanian J, Colau MM, Benhamou MA, Revel M. Validity, reliability, and responsiveness of the fingertip-to-floor test. Arch Phys Med Rehabil. 2001 Nov;82(11):1566-70. doi: 10.1053/apmr.2001.26064. PMID: 11689977.

Wachi M, Jiroumaru T, Satonaka A, Ikeya M, Noguchi S, Suzuki M, Hyodo Y, Oka Y, Fujikawa T. Effects of capacitive and resistive electric transfer therapy on pain and lumbar muscle stiffness and activity in patients with chronic low back pain. J Phys Ther Sci. 2022 May;34(5):400-403. doi: 10.1589/jpts.34.400. Epub 2022 May 1. PMID: 35527841; PMCID: PMC9057676.

Round 2

Reviewer 2 Report

All comments have been considered.